# Patterns of overlapping habitat use of juvenile white shark and human recreational water users along southern California beaches

**Patrick T. Rex**[ID]*[◯], **Jack H. May, III**[‡], **Erin K. Pierce**[‡], **Christopher G. Lowe**[ID][◯]

Department of Biological Sciences, California State University Long Beach, Long Beach, California, United States of America

◯ These authors contributed equally to this work.
‡ These authors also contributed equally to this work.
* Patrick.Rex@csulb.edu

**Data Availability Statement:** All relevant data related to this study are publicly available in the OSF repository (https://doi.org/10.17605/OSF.IO/ESVJQ).

## Abstract

Juvenile white sharks (JWS) of the Northeastern Pacific population are present in nearshore southern California waters and form mixed size class (~1.5–3 m) aggregations for weeks to months, often within 500 m of shore. These nearshore beach habitats are heavily used for human recreation (e.g., surfing, swimming, body boarding, wading, and standup paddle-boarding) and the amount of spatio-temporal overlap between JWS and humans is currently unknown. Increases in human population and the Northeastern Pacific population of white sharks have raised concern over human beach safety. To determine spatio-temporal JWS-human overlap at various spatial scales (e.g., across the entire southern California coast-line, across different distances from shore, and within specific beach locations), 26 beach locations across southern California were surveyed monthly resulting in 1644 aerial drone surveys between January 2019 to March 2021. Thirteen environmental variables were assessed to predict when spatio-temporal overlap between JWS and water users was high-est. Coast-wide distribution of JWS was clumped, limiting human-shark co-occurrence to specific locations, with 1096 of 1204 JWS observations occurring at Carpinteria and Del Mar Beach locations. Nearshore distribution indicated JWS are often close enough to the wave break to interact with some water users (median = 101 m, range = 2–702 m), although JWS had the most spatial overlap with stand-up paddlers. Daily human-shark co-occurrence was 97% at beaches where JWS aggregations had formed, and human activity showed high spatial overlap at shark aggregation sites. Although there is higher seasonal human-shark spatio-temporal overlap where aggregations form in southern California, the number of unprovoked shark bites across southern California is extremely low. This study provides evidence that high human-shark spatio-temporal overlap does not lead to an increased bite frequency in southern California, and there are a number of possible explanations as to why JWS are not biting water users despite daily encounters.

**Funding:** This study was primarily funded by the State of California Shark Beach Safety Program. PR also received partial funding from the Dr. Kenneth H. Coale Graduate Scholar award, awarded by the CSU Council on Ocean Affairs, Science and Technology. PR also received partial funding from the Southern California Tuna Club Marine Biology Educational Scholarship Foundation Graduate Research Grant from the Southern California Tuna Club. The funders had no role in study design, data collection and analysis, decision to publish, or preparation of the manuscript.

**Competing interests:** The authors have declared that no competing interests exist.

## Introduction

Many large predatory sharks, particularly tiger sharks, *Galeocerdo cuvier*, bull sharks, *Carcharinus leucas*, and white sharks, *Carcharodon carcharias* have been involved in unprovoked shark bites on humans, and all have experienced population declines since the 1970s due to overfishing and loss of key prey [1–8]. However, better fisheries management and protection efforts have enabled some of these populations to recover [9–12]. In addition, growing popularity of human ocean recreation (e.g., surfing, swimming, standup paddle boarding, body boarding, surf fishing, kayaking), exponential human population increase, and climate change has increased ocean use, especially along coastal beaches, which has increased the likelihood of shark-human encounters. Unlike terrestrial human-predator incidents, shark bites often receive significantly greater news coverage and tend to over-dramatize the dangerous nature of sharks [13,14]. As a result, public perceived risk of these low-frequency, occasionally lethal events is much higher than their actual occurrence, which has led to a lack of conservation efforts due to a negative public perception of sharks in general [15–17]. One of the potential reasons for this high perceived public risk is that there is very little data for natural spatio-temporal co-occurrence or encounter rates between humans and sharks. Areas with high human-shark co-occurrence are assumed to have higher encounter rates and potentially, higher bite rates. Previous estimates of shark-human encounter rates have been made using proxies such as beach counts of people, which do not provide actual measures of human water use, and likely overestimate encounter rates [23]. Human water user-shark encounter rates have not been measured, particularly across a large geographic area.

Most previous human-shark encounter studies assessed the effects of ecotourism on sharks, which involves providing food incentive to attract animals, the use of cages, or specifically seeking out sharks to create an encounter [18–21]. These forms of shark-human encounters through ecotourism are designed to increase encounter rates using food motivation or placing humans next to free-swimming sharks. Other studies focus specifically on retroactively assessing shark bite locations to identify spatio-temporal trends but lack adequate human water activity data to address actual encounter rates [22–24]. Only three studies to date have quantified unincentivized human-shark co-occurrence; however, these studies occurred at locations where there were unusually higher frequencies of shark bites on human water users [25–27]. Since unprovoked shark bites constitute an extremely small, but unquantified proportion of human-shark interactions, this creates a large gap in knowledge particularly for estimating shark bite risk. Specifically, it creates concerns about how conservation will impact beach safety for species like white sharks that use coastal, heavily human-populated habitats with high interannual frequency [28].

Juvenile white sharks (JWS) of the Northeastern Pacific (NEP) population utilize the heavily populated southern California coastal beach areas as nursery habitat [9,28–33] and may have high co-occurrence with people. Furthermore, the NEP population of white sharks has likely been increasing over the last 20 years [9,11,31,34], so human-shark co-occurrence (overlapping habitat use) may be increasing as both human and shark populations increase. JWS form mixed size-class aggregations (> 40 individuals), ranging from 1.5–3 m (TL), and aggregate for weeks to months at popular beach recreation areas in California [33,35]. This presents a potential bite risk due to the higher encounter rates as white sharks 2.5–3.5 m TL are generally the size class involved in bites on humans in southern California [36,37]. Thus, there is increased concern for public safety as southern California can have ~129 million beach-specific visitations per year and has likely increased due to climate change and rising popularity of beach related water activities [38,39]. Furthermore, there is very little data on how many water users (e.g., surfers, bodyboarders, swimmers, waders, standup paddleboarders) use these

nearshore waters (< 500 m from the shoreline, < 8 m water depth), how types of water recreation activities vary among beaches, and their proximity to the shoreline.

Aerial surveys have proven to be effective methods for quantifying human beach use [26,27,40–42] but are relatively ineffective at observing sharks when they are below 2.5 m of water [43,44]. However, JWS are often near the surface (< 2.5 m [28,45]) and can be surface associated for up to 70% of the day [35]. These behaviors make aerial surveys particularly effective at monitoring white shark nearshore behavior [46–50]. Unmanned Aerial Vehicles (UAV), or drones, are a cost-effective, easy-to-use, publicly available tool that may complement ecological studies by providing high-resolution, georeferenced video. Furthermore, drones can operate at lower altitudes (< 120 m) and lower speeds than piloted aerial surveys which make for accurate nearshore observations [51]. This study represents the first comprehensive study of shark-human co-occurrence across a large, heavily populated coastline using an emerging ecological tool.

## Methodology

### Study site

The study region spanned from Santa Barbara (34.420830,-119.698189) to San Diego (32.715736,-117.161087), encompassing all five southern California coastal counties and the known aggregation areas for JWS [28,35] (Fig 1). Aerial surveys were conducted routinely at 26 beach locations that had unrestricted airspace as defined by United States Federal Aviation Administration regulations [52]. Surveys were flown at each beach once a month unless environmental conditions prohibited safe flights. Conditions that restricted surveys included low clouds (< 182 m) or high wind (> 8 m/s) in accordance with FAA regulations and drone flight speed limits. Surveys were conducted from January 2019 until March 2021, spanning a total of 26 months.

Federal Aviation Administration part 107.41 airspace authorizations and certificates of authorization were acquired to fly in Seal Beach (WSA 19029) and North Beach, Coronado (WSA 17090). All other survey locations were in unrestricted airspace, or upon request from the city or beach safety department managing the beach location.

An Institutional Review Board (IRB) and protection of human subjects permit was not required as humans subjects were observed at altitudes that prevented facial identification of individuals. All JWS observations were conducted under State and federal permits, and under California State University Long Beach's Institutional Animal Care and Use Committee (IACUC) protocol #364. No JWS were disturbed or handled within this study.

### Aerial surveys

Aerial surveys were conducted primarily with a DJI Phantom 4 Pro v2.0 (Da-jiang Innovations) quadcopter. Orthogonal video was recorded in 4k resolution (3840 x 2160), 30 fps, at 16:9 aspect ratio. Recorded metadata included an on-board barometer, high-resolution GPS unit (± 10 cm based on manufacturer specifications), and on-board inertial measurement unit, providing date, time, frame associated GPS location, altitude, and home location. The altitude above the water of the drone during surveys varied from 60 to 120 m. Altitude varied to ensure that the shoreline, wave break, and all potential human subjects were within the same frame of the drone. Shoreline was determined by where the dry sand interfaced with the lapping of waves. Wave break was defined as the location proximate to the shoreline where the waves were physically breaking. If breaking waves were not visible within each image, the farthest offshore edge of the surf whitewash was used as a proxy. Surveys were flown manually at 5.5–6.0 m s$^{-1}$ following the curvature of the shoreline until the drone was 1 km away from the survey

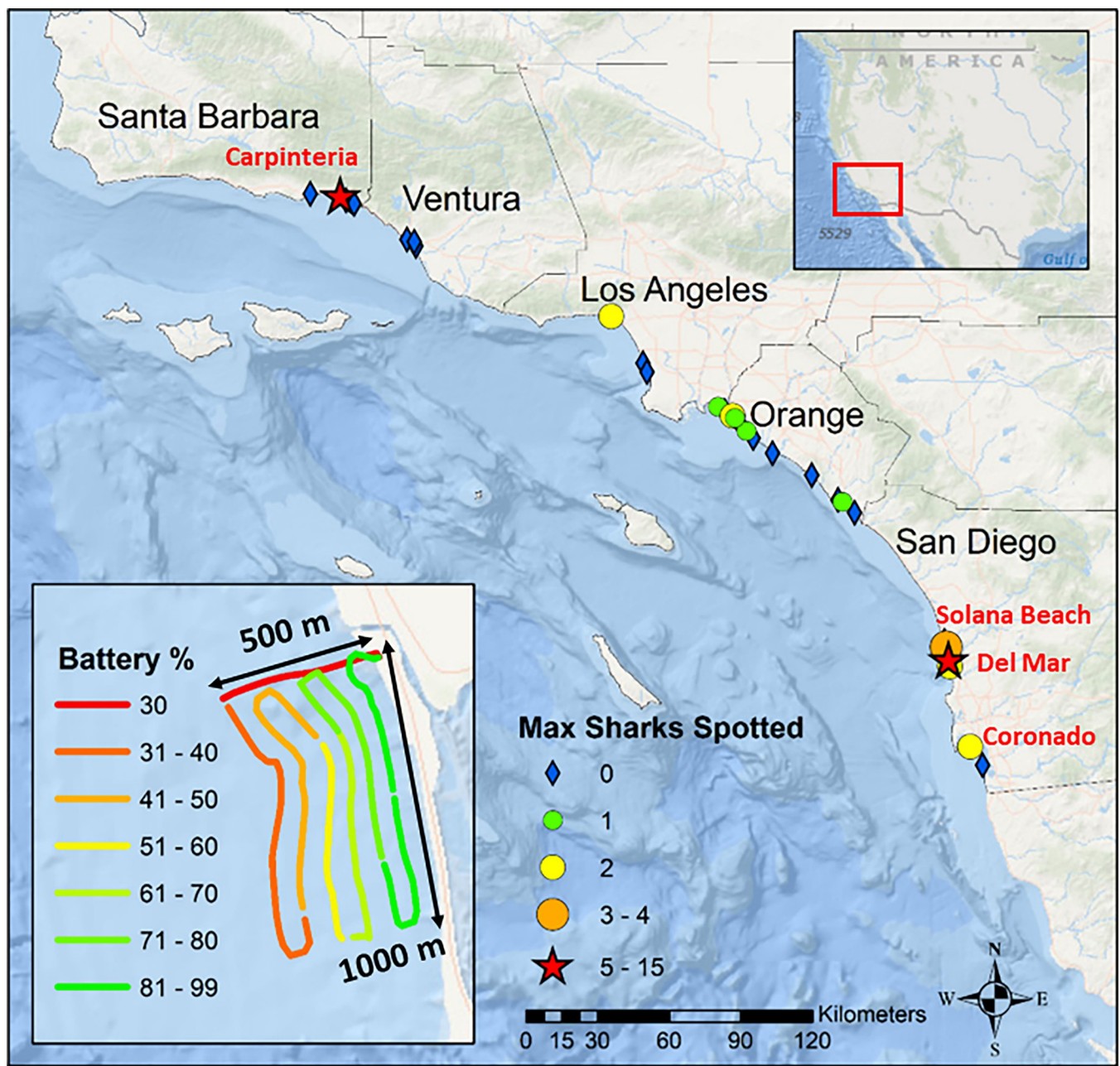

**Fig 1. Map of overall survey area, beaches surveyed, and the maximum number of sharks observed during a single survey.** Beaches labeled with red text were aggregation locations where 3 distinct individual sharks were observed on consecutive survey days. Our definition for aggregation site was limited to locations where 5 distinct individual sharks were observed on consecutive survey days and is limited to Carpinteria and Del Mar (stars). Map inset is an example of drone survey path. Base map and map data were produced in ArcGIS. Base map data produced by Esri, Garmin, the General Bathymetric Chart of the Oceans, and the National Oceanographic and Atmospheric Administration National Geophysical Data Center. Map imagery from USGS can be viewed here: https://apps.nationalmap.gov/viewer/.

starting position (Fig 1 inset). Once at 1 km from its starting position, the survey continued in a "lawnmower" pattern, following the curvature of the shoreline, until the drone was 500 m offshore. (Fig 1 inset). All subsequent paths of the survey after the first transect were flown at 60 m, which provided a consistent, optimal field of view allowing for adequate identification of observed sharks to species. Area covered within the drone's frame of view ranged from 6,420

m$^2$ (60 m x 107 m) to 25,833 m$^2$ (121 m x 214 m) depending on drone altitude (60 m and 120 m respectively) with the maximum area covered within a single flight survey of 0.7 km$^2$.

If a JWS was spotted, the drone was positioned directly above the shark to get an accurate GPS location of the shark's location. Transects were stopped to observe the shark's behavior and observe for potential human-shark interactions. Although high-resolution tracks of the animal were collected, to keep human and shark surveys comparable, only the first GPS record for each individual shark observation was used in the analysis in this study. If JWS were observed at a beach, repeated surveys were flown at that beach location until the JWS was no longer observed or no more batteries were available for flights. This was to increase the dataset for JWS distance from wave break, and to potentially observe as many individual JWS as possible within a single survey day. This did create a discrepancy in effort among beach locations; however, to keep human data comparable, the first transect that encompassed human area use was only performed once per location per survey day. Furthermore, JWS positions were opportunistically included in the dataset when they were observed during acoustic tagging (external dart-tagging) operations by California State University Long Beach scientists to increase the JWS observations. The GPS point was only included upon first observation to exclude any additional shark movement that may have been caused during tagging operations. Human subjects were not surveyed during these operations to keep effort comparable between locations. Surveyed beaches varied in length (2–7 km long), so multiple flights were required to fully cover the entire beach; beach survey areas ranged from 0.5 to 3.5 km$^2$.

## Survey processing

To quantify spatio-temporal overlap between humans and sharks within the nearshore, the distance of every observed subject (human or shark) was measured perpendicular to the wave break line. Distances were measured by extracting an image from recorded surveys every 5 sec. Image overlap ranged from 32%-93% depending on drone altitudes (60–120 m), representing the minimum and maximum survey altitude (S5 Table). The overlap between images was high to increase count accuracy and reduce the potential for missing sharks. To reduce potential double measurement of observed human subjects, distances and GPS locations of human subjects were only quantified for the first frame they were encountered. They were easily tracked between frames based on unique identifiers, e.g., size, clothing color, board shape and color, etc, to ensure they were only included in the dataset once. Humans observed during surveys were counted and categorized into 5 separate groups: waders, swimmers, bodyboarders, surfers, and stand-up paddlers (SUPs). Waders were defined as people walking/standing along the shore up to waist deep water, not using a board of any kind. Swimmers were identified as water users fully emersed in the water and not standing. These groups were selected because they are the most popular recreational beach-related activities in southern California with the potential to be near JWS within surveys.

Distances from the wave break to subjects were measured using ImageJ (NIH, v.1.8.0). ImageJ was calibrated by converting pixel length within the image to a cm/pixel length using Pix4D ground sample distance calculator (https://support.pix4d.com/hc/en-us/articles/202560249-TOOLS-GSD-calculator). Conversion is based on the DJI Phantom 4 Pro v.2 drone's focal length (8.8 mm), sensor width (13.2 mm), sensor height (8.8 mm), flight altitude, image height, and width in pixels. Altitude was determined directly from the drone's metadata embedded in recorded footage from the survey. Drone camera parameters (e.g., focal length, sensor dimensions) were taken directly from DJI manufacturer's specifications. Image dimensions were 3840 x 2160 pixels. Subjects between the wave break and shoreline were indicated by negative distance measurements and subjects outside the wave break were indicated by

positive distance measurements, using the location of wave break as a standardized physical feature.

To measure subject distance from the wave break when the shoreline or wave break was not in view during the survey path, wave break shapefiles were created and georeferenced in Arc-Map (Esri, v.10.8). These were created by georeferencing sequential images from the first leg of the survey transects that follow the coastline with both shore and wave break in the frame. Shapefiles were created by tracing the prominent wave break, or edge of the whitewash as a proxy for the wave break within frames using the "polyline" tool within ArcMap. This created a spatio-temporally accurate representation of the wave break at the point when subjects were spotted from which the distance of the subject to the wave break could be measured. Distances were measured from the GPS location of the shark when it was spotted to the wave break shapefiles using the Nearest tool in ArcMap (Spatial Analyst package). GPS locations were recorded when the drone was directly over observed sharks to attain the most accurate positioning at the time of observation since the GPS unit is within the center of the drone.

Spatial resolution and distributions of both people and sharks within this study varied in scale. Larger-scale distribution patterns of sharks and people was defined as "coast-wide" distributions. Distributions of humans and sharks in proximity to the wave break and shoreline was defined as "nearshore" distributions. Lastly, beach-specific area use and spatial overlap was defined as "beach-specific" distributions.

## Environmental data acquisition

A wide variety of environmental variables were compared to determine if they were good predictors of water user activity or shark presence at each beach (S1 Table). Wave direction, wave height, and mean as well as peak wave period were collected from local weather buoy databases maintained by the Coastal Data Information Program (CDIP) [53]. CDIP buoys are spaced every ~10 km along the southern California coastline with variables measured hourly. Air temperature, sea surface temperature (SST), SST anomaly, wind speed, and wind direction were pulled from 0.1˚x0.1˚ to 0.5˚x0.5˚ spatially gridded, modeled databases maintained by NOAA [54]. Measured daily lunar illumination was determined by accessing the National Aeronautics and Space Administration Moon Phase and Libration Database for the Pacific Standard Time zone [55]. Total Cloud cover, cloud base height, and UV B radiation were pulled from European Centre for Medium Range Weather Forecast reanalysis on 0.25˚x 0.25˚ spatially gridded modeled data (S1 Table).

## Data analysis

Human and shark distance from wave break data across all subject groups were non-normal and could not be normalized to meet assumptions of a General Linear Model, so a Kruskal-Wallis test (rstatix package [56]) was used to determine differences in distributions, followed by a Dunn's test with a Bonferroni p-value correction to make pairwise comparisons between each water user group and sharks using the rstatix package [56] using R Studio (V.4.0, R Core Team). Mean relative abundance for each beach location was determined across the entire study period to assess how overall human-shark relative abundance may differ between study locations. Subjects were separated by group, summed across all individual surveys conducted, averaged by the number of surveys conducted, and then a relative abundance was determined based on the mean number of observations of each group per survey. This was repeated for every beach surveyed. Seasonal abundance was limited to aggregation sites only in separate analysis, as these were the only sites that had shark observations across multiple seasons. Maximum shark density was also calculated based on a conservative estimate of the number of

individual sharks observed in a survey per day. Since beaches were different in size, densities were normalized by taking the maximum number of confirmed individual sharks observed by survey and dividing this by beach area surveyed. Geospatial distribution overlap between observed humans and sharks within the two largest JWS aggregation beaches, Carpinteria and Del Mar, was determined using kernel density estimations using the MASS package [57]. GPS locations of where both sharks and humans were encountered within surveys across the entirety of the study were compared. Human-shark co-occurrence was defined as whenever sharks were spotted at the same beach within the same survey day. Survey day was used as the base unit of measure for co-occurrence because it has been shown that JWS have high site fidelity within aggregations [28]. White sharks are also known to "patrol" beach areas in the nearshore [49]. Therefore, there is a high probability that sharks co-occurring at the same beach will likely encounter people also using the same habitat throughout the day. Furthermore, co-occurrence was observed during tagging operations when JWS swam near a human subject, even if human subjects were not surveyed during these operations. Events where sharks were in close proximity, e.g., within 100 m of a human, were opportunistically observed. However, much finer resolution methodology is required to accurately analyze these events and thus these data remain outside the scope of this study.

To determine which environmental variables might best predict the presence of humans and sharks, a combined forward and backward General Additive Mixed Model (GAMM) was run for each subject group using the gam package [58]. Each step-wise model had every individual subject group's abundance as the response variable and included all variables as predictor variables, excluding any less relevant ones (e.g., air temperature for JWS). To further reduce model complexity, variables that were determined significant via step-wise GAMM were ranked by importance (caret package, [59]). A correlation matrix was also used to assess significant correlation between environmental variables ($p < 0.05$). If two or more variables were correlated and potentially redundant, e.g., peak wave period and mean wave period, the variable that had a higher importance score was kept in the model. The exception to this was air temperature, sea surface temperature, and month. While month and both sea surface and air temperature are correlated, local thermal anomalies such as upwelling events or heat waves could affect the abundance of people or sharks within the nearshore area despite the time of year. Furthermore, Spurgeon et al. [60] observed that anomalous decreases in temperature may cause JWS to emigrate from aggregation sites. As a result, SST anomaly was not deemed redundant.

## Results

### Human-shark distribution overlap

In total, 1644 individual surveys across 163 survey days were conducted from January 2019 through March 2021 resulting in over 700 hrs of survey video. Across all surveys, waders ($n = 9068$) and surfers ($n = 8033$) were the most abundant water users, with swimmers ($n = 595$) and bodyboarders ($n = 1296$) intermediate, and SUP ($n = 181$) among the rarest observed group. Waders and surfers were ubiquitous across all surveyed beach locations and all seasons. Bodyboarders and swimmers were observed at almost every beach location but were observed less frequently outside of summer months (June-August). Stand-up paddlers were rarely observed and generally only in two locations, Carpinteria or Del Mar. A total of 1204 JWS were observed during the study period, with a majority observed at two sites with low coast-wide abundance (Figs 1 and 2). Sharks were only counted as individual sharks within the study if they could, without question, be identified as distinct individuals, e.g., multiple sharks in the same frame of view of the shark, distinct scarring, visible size differences,

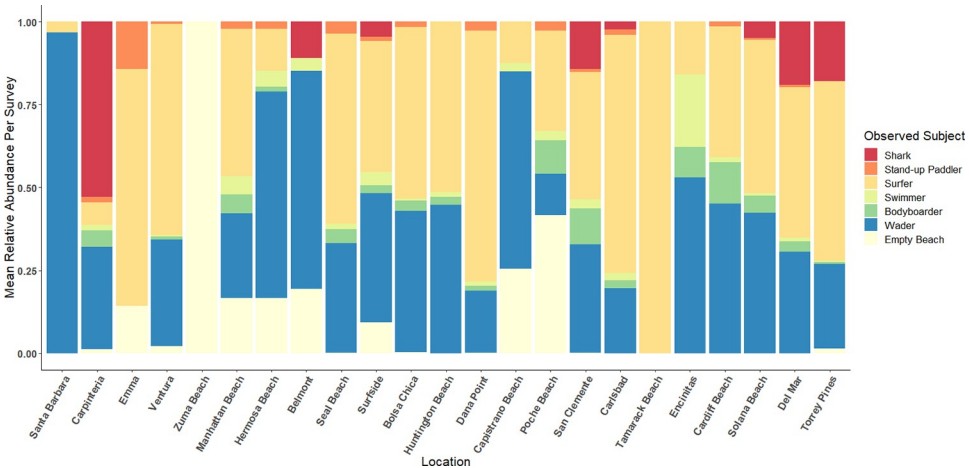

**Fig 2. The relative abundance of subjects for each survey site across the entirety of the study.** Individual beaches are organized from the most northern site, Santa Barbara, to the most southern site, Torrey Pines. "Empty Beach" indicates surveys where no humans were observed. Some sites that were within 5 km of each other geographically were combined.

unique external tag placement, etc. Otherwise repeat observations of JWS within the same survey were assumed to be the same individual.

In general, shark sightings were rare from January 2019 through October 2019. Only 1–2 JWS were observed off Surfside and Capistrano Beach (Orange County), Belmont Shore (Los Angeles County), and Solana Beach (San Diego County) per survey, from February 2019 through August 2019 (Figs 1 and 2). JWS were not observed again at any of these beaches for the remainder of the study. Larger aggregations were first observed in August 2019, with up to five individual sharks sighted within one survey day in Carpinteria. This aggregation also showed the highest max density of sharks with 15 individuals being sighted in a single survey (0.086 sharks/ha) and up to 90 JWS observations per survey day. A second, persistent aggregation formed in Del Mar in June 2020, where sharks were consistently sighted until March 2021. This second aggregation had a maximum recorded density of 11 individual sharks observed in a survey day (0.055 sharks/ha) and up to 66 shark observations in a single day. Shark aggregations and general sightings were geographically discrete; despite apparent similarities in habitat, sharks were seldom sighted at neighboring adjacent beaches to aggregation sites. Sharks were significantly closer to the wave break when observed within aggregations (median distance from wave break = 82 m) across all months than when they were observed at beaches outside of aggregation sites (median = 216 m) (Wilcoxon Rank Sum, $W$ = 46931, $p < 2.2$x$10^{-16}$). However, the number of shark observations was much lower during Winter months (December through February). This ranged from 1–6 (median = 2) sharks per survey day at Carpinteria and 0–8 (median = 3) sharks per survey day in Del Mar respectively. Summer months (July through August) had the highest number of observations within Carpinteria with 1–90 (median = 18) shark observations per survey day. However, Del Mar had the highest number of observations within Fall months (September through November), with 1–66 (median = 18) shark observations in Del Mar per survey day.

While there was considerable overlap in nearshore distribution of water users from the shoreline to outside the wave break, certain groups were more likely to be distributed outside the wave break and closer to shark nearshore distributions (Fig 3). Waders and bodyboarders were rarely seen outside the wave break and share the least nearshore distribution overlap with sharks (Fig 3). Surfers and swimmers occupy the same area in the nearshore and have

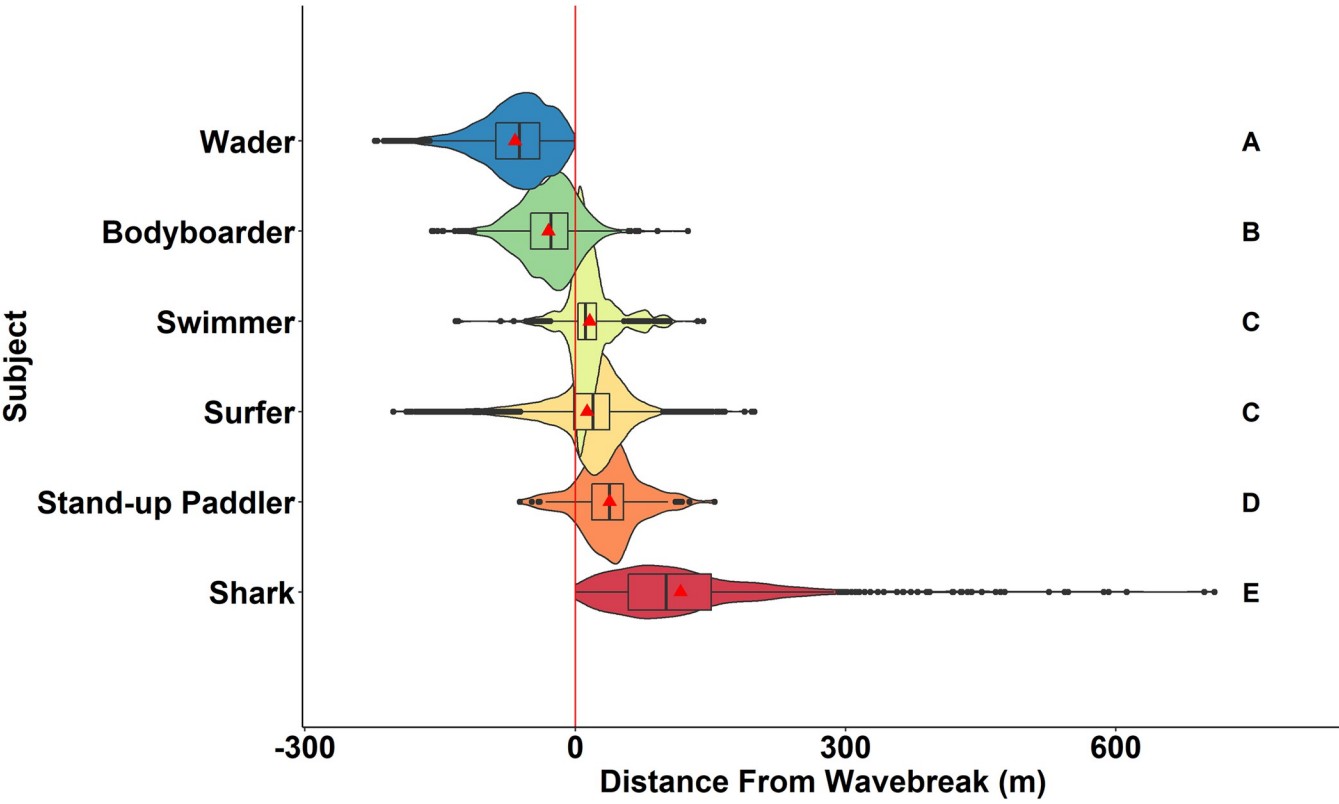

**Fig 3. The distance of subjects from wave break observed in surveys showing nearshore distributions.** The red line indicates the wave break. Negative values indicate the subject was between the wave break and shoreline (wave wash), and positive values indicate position offshore of the wave break. Letters indicate significantly different distributions based on pairwise comparisons.

substantial overlap with those sharks that are closest to the wave break (Fig 3). SUPs were generally farthest outside the wave break and share the most nearshore overlap with JWS, although they were least common in occurrence (Fig 3). JWS ranged in nearshore distribution from 2–709 m offshore but were observed routinely within encounter distance of human water users (median = 101 m) (Fig 3).

Co-occurrences of sharks and humans were rare outside of the two main aggregation sites of Carpinteria and Del Mar. Human-JWS co-occurrences were observed during 74 survey days out of 163 survey days, and 58 (78.4%) of those were at aggregation sites. Of the 66 survey days conducted at aggregation sites, 58 resulted in observed co-occurrence of JWS and water users. Of the 8 out of 66 survey days without co-occurrence, six were the result of not observing sharks, and only two were the result of not observing humans. Aggregation sites had an observed proportion of 97% (58/60) daily human-JWS co-occurrence. The remaining survey days where co-occurrences were not observed were in December and January when JWS abundance was lower [28].

Although co-occurrence, in general, was highest at aggregation sites, there were differences in the coast-wide relative abundances of water users. Although relative abundances were low, SUP nearshore distributions overlapped the most with that of JWS (Fig 3). However, an order of magnitude more surfers ($n = 8033$) were observed coast-wide than SUPs ($n = 181$). Surfers on average comprised of 40 ± 27% relative abundance of all water users observed in individual coast-wide surveys (Fig 2). In comparison, SUPs accounted for only 1.6 ± 2.9% of survey composition within coast-wide relative abundance (Fig 2). Surfers were also observed in higher

relative abundance across the entire study period within Carpinteria and Del Mar (Fig 2). Surfers, particularly those in the "line-up" outside the wave break were observed to have a much higher rate of co-occurrence with JWS. Surfers co-occurred with JWS in 54 out of 74 survey days (73%) with an average of 45 surfers observed when JWS were observed at that beach. SUP, in contrast, only co-occurred 38% of the time (28 out of 74 survey days) with an average of 2 SUP observed at the same time as JWS. Only swimmers had lower co-occurrence, with a rate of 34% (25 out of 74 survey days), although there was an average of 8 swimmers observed during survey where co-occurrence was observed. Bodyboarders also had relatively high co-occurrence with JWS at beaches, with 51% observed rate (38 out of 74 survey days) and an average of 14 bodyboarders observed during surveys when JWS were observed. Although surfers were the most likely to be in proximity of JWS due to their higher abundance and distribution (Fig 2), waders had the highest co-occurrence (88%, 65 out of 74 survey days). However, waders were always observed between the wave break and shoreline with zero nearshore distribution overlap with JWS.

## Environmental effect on presence in the Nearshore

Step-wise, mean-anchored GAMMs identified environmental variables that were significant predictors for the abundance of each group (water users and sharks) (Table 1). For example, board sport abundance increased by the wave period; surfers and SUP abundance increased when mean wave period was longer than 10 sec, while bodyboarders had the highest abundance when mean wave period less than 6 sec (Fig 4). Higher wind speeds showed a decreased abundance for most water user groups, which was constrained by drone flight capabilities (wind speeds < 8 m/s). There was a strong seasonality across all water user groups with higher abundance throughout the summer and decreased abundance during the winter. As a result, SST, air temperature, and month, which are correlated, were significant predictors of abundance for all groups (Table 1). In general, abundance for all groups was highest during the summer with high UV, high SST, and moderate wave height (~1 m) (Table 1).

   Shark abundance was predicted by several environmental variables (Table 1). Seasonal variables such as month, SST were the best predictors for shark presence. Wave variables such as wave height and wave period were also predictors of shark abundance (Fig 4), with the highest abundance when wave height was <1 m high and period of 5–8 sec. High UV was also correlated with an increase in shark observations within surveyed areas. However, observations of

**Table 1. The product of Step-Wise GAMM analysis of environmental parameters.** Plus signs indicate a positive relationship, minus signs indicate an inverse relationship, "p" indicates a negative parabolic relationship, and "q" indicates a quadratic function with no single peak. Bolded variables are common across all subject groups.

| Subject | Step-wise Product |
|---|---|
| **Shark** | presence~s(Tidal Range$^+$)+s(Sea Surface Temperature$^q$)+s(Lunar Illumination$^q$)+s(Wave Height$^P$)+s(Wind Speed$^P$)+s(Sea Surface Temperature Anomaly$^-$)+s(Wave Direction$^q$)+s(Location$^q$)+s(Mean Wave Period$^P$)+s(UV B$^+$)+s(**Month**$^q$)+s(Day$^P$) |
| **SUP** | presence~s(Tidal Range $^+$)+Air Temperature$^+$ +Wave Height $^P$+s(Wind Speed $^-$)+s(SSTAnom $^P$)+s(Mean Wave Period $^P$)+UV B$^-$ +**Month** $^P$ |
| **Surfer** | presence~s(tidal range$^P$)+Sea Surface Temperature$^+$ +s(Wave Height$^P$)+Wind Speed$^-$ +s(mean wave period$^P$)+s(UV B$^P$)+s(**Month**$^P$)+s(Day$^P$)+s(Location)+s(hour $^P$) |
| **Bodyboarder** | presence~s(Air Temperature$^+$)+s(Sea Surface Temperature $^+$)+Wave Height$^+$+s(wind speed$^-$)+s(SSTAnom$^P$)+s(WaveDir$^P$)+Mean Wave Period$^+$+s(UV B$^+$)+s(Day$^+$)+s(**Month**$^P$) |
| **Swimmer** | presence~Air Temperature$^+$ +Sea Surface Temperature$^+$ +s(Wind Direction)+SSTAnom$^{P, 0}$ +Peak Wave Period$^+$ +UV B$^+$ +**Month**$^{P, Jun}$ |
| **wader** | presence~Air Temperature$^+$ +Sea Surface Temperature$^+$ s(Wave Height$^P$)+Wind Speed$^-$ +s(Mean Wave Period) +Cloud Base Height$^+$ +Day of the Week$^+$ +s(Location)+s(**Month**$^P$) |

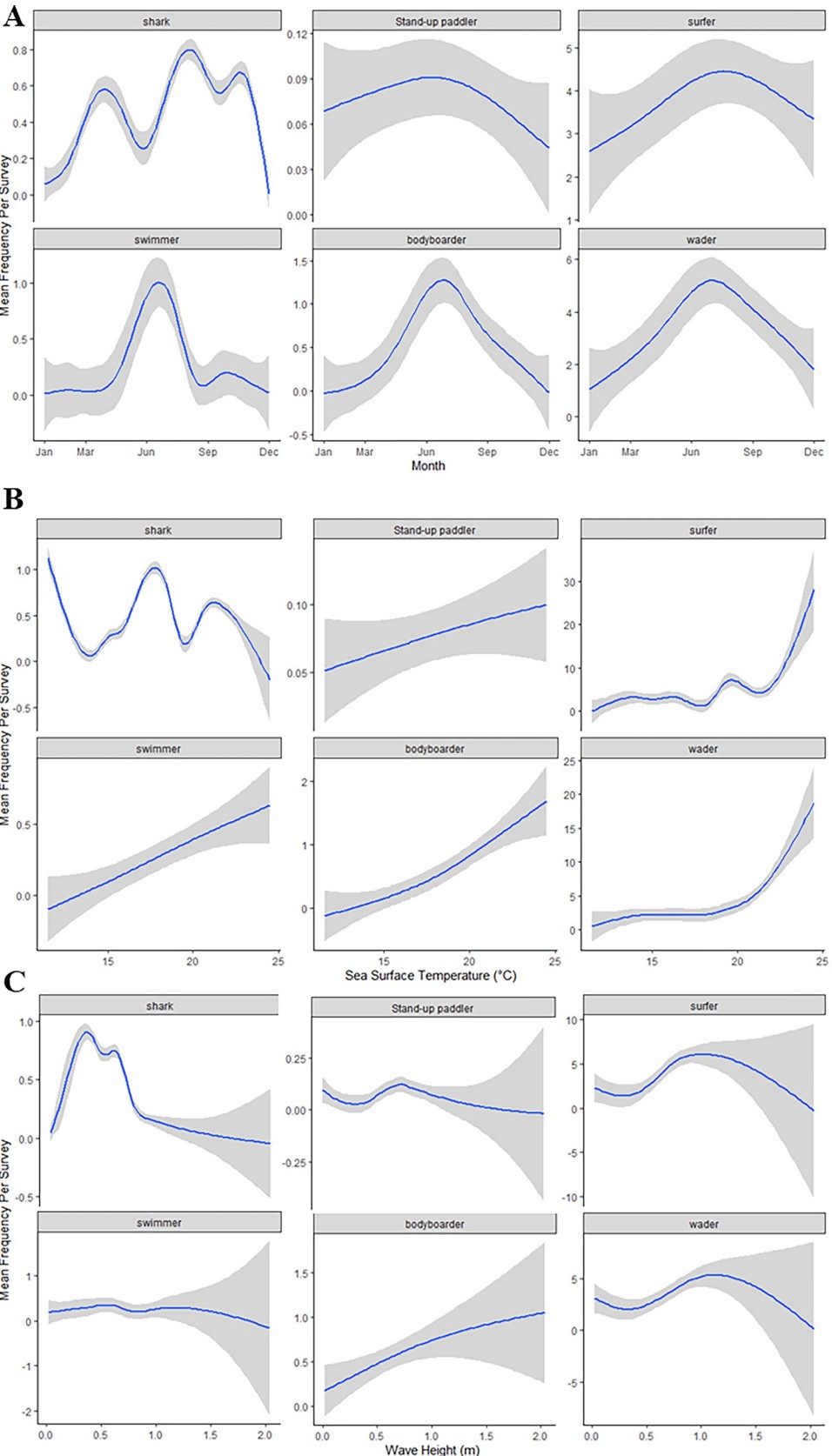

**Fig 4. Mean anchored General Additive Mixed Model plots of three environmental variables by subject.** Y Axes vary for each group due to the difference in observations for each group per survey. A.) The predicted average number of JWS and human water users within southern California based on month of the year. B.) The predicted average number of JWS and human water users within southern California based on SST. C.) The predicted average number of JWS and human water users within southern California based on wave height.

JWS with drones were limited to more fair-weather conditions, e.g., sunny with high UV, high cloud base height which decreases cloud glare, and low total cloud cover for increased contrast between clear water and dark shark bodies. While total cloud cover and cloud base height were not statistically significant, more overcast conditions made spotting JWS more difficult due to poor optical properties. UV and total cloud cover were significantly correlated, $p < 0.05$ via correlation matrix, so UV may be a proxy for total cloud cover. However, due to Federal Aviation Administration regulations, flights were limited to high cloud base height, (above 610 m), which may account for the lack of statistical significance. Furthermore, while JWS were primarily observed only at the two aggregation sites, individuals were only occasionally seen at other surveyed beaches throughout the study period. Due to this, location was not a significant predictor of shark presence.

## Human-JWS spatio-temporal overlap within aggregations

Carpinteria represented one of the two large aggregation sites within southern California observed during this study. The aggregation persisted from August 2019 through the end of the study in March 2021 (Fig 5B). Although shark observations decreased during the winter, shark observations were consistent across all other seasons and peaked during the summer. Human water user observations, in contrast, were low until the summer and consisted mainly of waders along the shore with a peak abundance of all water user groups during summer months (Fig 5B). As a result, this beach is classified as a "low-use" beach area by water users. June through August were the months of highest overlap between humans and JWS (Fig 5B). While JWS were observed along the entire 3.5 km stretch of beach most sightings were clustered at a single area in the center of the survey area (Fig 5A). Human beach-specific space use was more contracted along a 627 m stretch of beach in between the only two public beach access locations (Fig 5A). Despite these differences in overall beach-wide distribution, the primary area of use for both human and JWS overlapped throughout the course of this study (Fig 5A). Nearshore distributions at Carpinteria Beach were significantly different than coast-wide trends likely due to JWS being closer to shore at this beach (Figs 3 and 5C and S4 Table for direct comparisons). While surfers and swimmers still occupied the same area, there was no significant difference between swimmers and SUPs, nor SUPs and sharks (Kruskal-Wallis and post-hoc Dunn's test; S3 Table). While sharks were routinely observed meters from the wave break, they were also observed up to 700 m outside the wave break at this location. Despite being a relatively low-use beach, Human-JWS, beach-specific co-occurrence was high at this beach where co-occurrences observed during 36 out of the 38 survey days (94.7%).

Del Mar was the second largest JWS aggregation in southern California observed during this study, which persisted from June 2020 through the end of the study in March of 2021. Spring 2020 data were limited due to difficulties sampling this beach site because of COVID-19 lockdowns which restricted all beach access within San Diego County. While the stretch of beach where JWS were observed at Del Mar is similar in size to that of Carpinteria Beach (~4 km long), Del Mar has more prominent patches of rocky reef throughout the nearshore area. Del Mar Beach was used more extensively throughout the year for human recreation than Carpinteria, with a larger, year-round surfing population due to the more consistent surf conditions (Fig 6B). Shark observations were also fewer than in Carpinteria for most of the year;

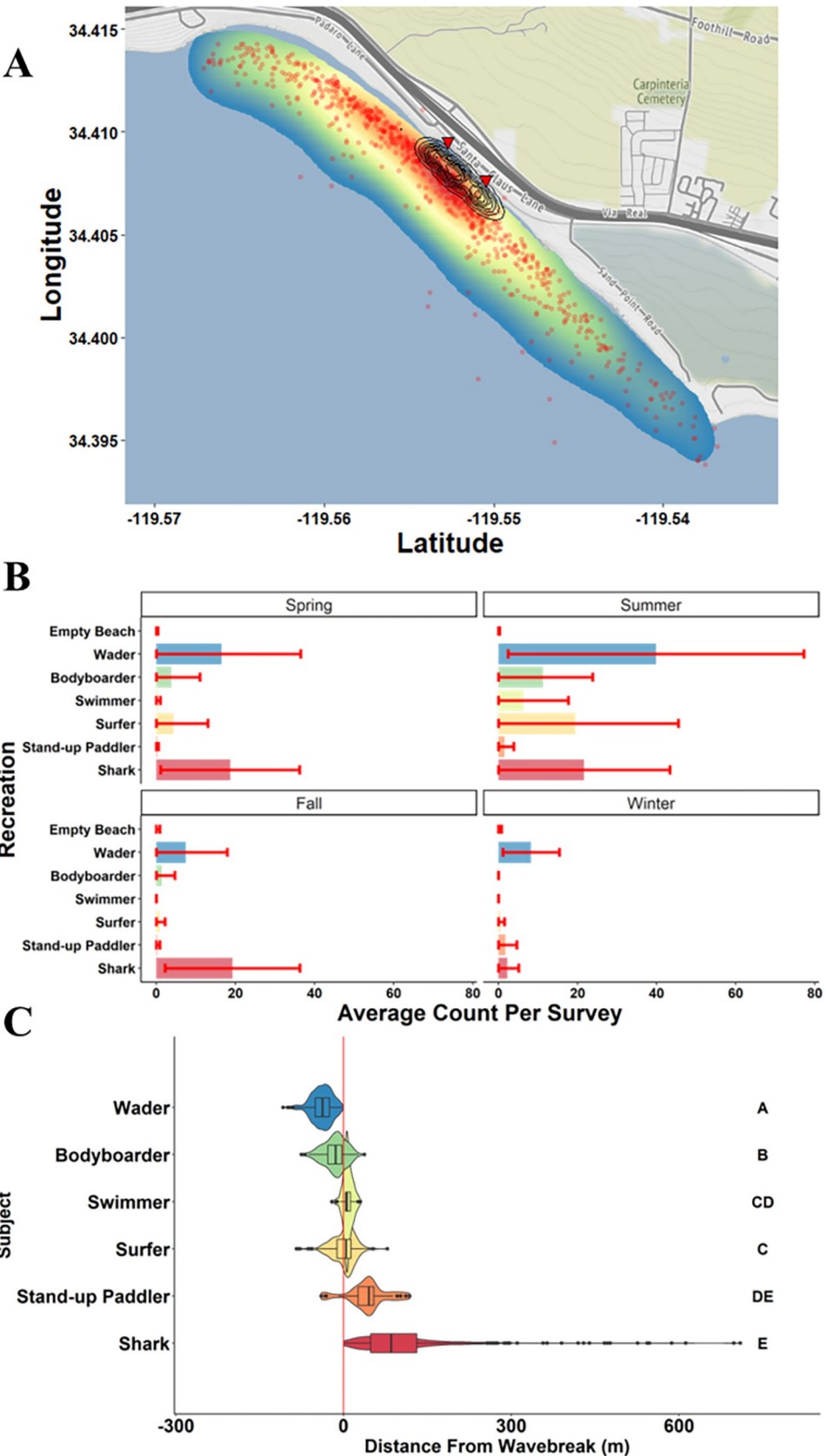

**Fig 5. Spatial distribution and abundance of water user and JWS at the Carpinteria aggregation site.** A.) depicts
the kernel density estimation (KDE) overlap between humans of all groups and sharks spotted across the entirety of the
study. Red dots represent GPS locations where individual sharks were sighted among all surveys. Black contour lines
are all human water user group kernel densities. Red Triangles indicate beach access locations. B.) Average (± SD)
number of observations of water users and JWS per survey by season. C.) Nearshore distances from wave break
distributions for water users and JWS, separate letters indicate significantly different distributions (see S3 Table for full
pair-wise comparison). Map and imagery data from Stamen Design open-sourced map tiles under the Creative
Commons Attribution (CC BY) with data from OpenStreetMap contributors. Map imagery can be found here: http://
maps.stamen.com/terrain/#14/34.3722/-119.4698.

however, fall months had the peak number of shark observations in Del Mar (Fig 6B). This
resulted in fall having the most temporal overlap between humans and sharks. Beach-specific
spatial overlap between humans and sharks at Del Mar was different than Carpinteria. While
sharks were still observed to use the entire beach area, there were two focal areas associate with
rocky reefs along this shoreline where JWS were observed at higher densities (Fig 6A). Human
beach-specific area use was also much larger than Carpinteria, spanning the entire beach,
which corresponded to more regularly spaced public beach access points along the beach (Fig
6A). Human core area-use also concentrated on the same rocky reefs where JWS were
observed, creating high beach-specific spatial overlap between JWS and humans (Fig 6A).
Contrary to the nearshore distribution observed across the entire study, Del Mar had more
observed nearshore distribution overlap among some subjects (Figs 3 and 6C). There were no
significant differences in nearshore distributions between surfers, swimmers, and SUP; how-
ever, only SUPs shared significant overlap with JWS (Fig 6C, S4 Table). While survey days
where sharks were observed were more limited in Del Mar than in Carpinteria, sharks and
water users were detected at Del Mar 18 of the 18 survey days (100%).

## Discussion

Despite their relatively low abundance, JWS showed a high spatio-temporal overlap with
humans only at beaches where nursery aggregations had formed. Aerial survey methods using
drones allowed for a cost-effective tool for quantifying human water user distribution along a
large, heavily populated coastline, providing the first large-scale survey of white shark/human
shared habitat use. Historically, aerial surveys have shown to be an effective tool for quantify-
ing human beach recreation use [26,27,40–42,61,62] and drones are being used to study shark
nearshore behavior with increased frequency [46,48,50,63–65]. However, characterization of
human water user and shark distributions across varying spatial scales is novel. Examining
human and shark survey data at various spatial scales, e.g., coast-wide, nearshore, and beach-
specific distributions, provides insight into the degree of encounter and estimates of risk. How-
ever, the use of drones is not allowed in many restricted airspaces and can thus lead to biases
in shark distributions and estimates of spatially explicit abundances and densities.

### Human distribution and Nearshore area use

Human coast-wide distribution was ubiquitous despite certain beaches being more popular
for recreational human use (Figs 2 and 7). Southern California's large population of 18 million
people [66] and $24 billion ocean-related industry [67] may account for high prevalence of rec-
reational water activities in beach areas across southern California. People may also use less
popular beaches, despite limited parking, smaller beach area, poorer surf conditions, and more
distant from highly populated areas, as beaches become more crowded, particularly during
peak months [68–70]. While it is well-documented that human beach use significantly
increases during Summer and holidays [38,41,42], an increase in human coast-wide

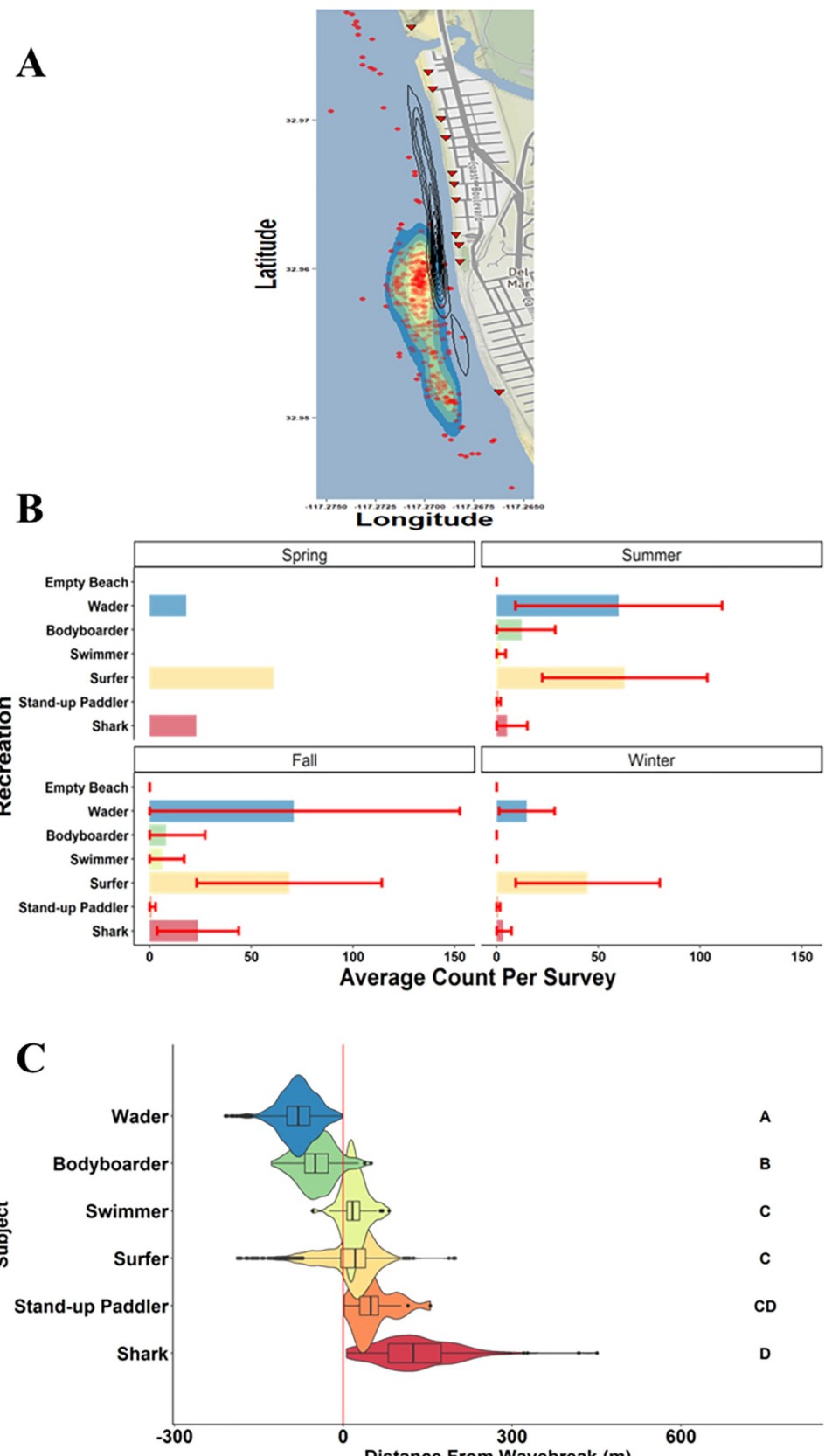

**Fig 6. Spatial distribution and abundance for human water user and JWS at the Del Mar aggregation site.** A.) 95% kernel density estimation (KDE) overlap between sharks and humans throughout the study. White contour lines represent human 95% KDE, and red triangles represent public beach access. B.) represents average observations of each group by season with associated standard deviation. C.) Del Mar specific distributions within the nearshore area. Different letters indicate significantly different distributions (see S4 Table for full pair-wise comparison results). Map and imagery data from Stamen Design open-sourced map tiles under the Creative Commons Attribution (CC BY) with data from OpenStreetMap contributors. Map imagery can be found here: http://maps.stamen.com/terrain/#12/32.9537/-117.2633.

abundance could be explained by both people choosing to use less popular beaches as well an as influx in tourist visitation at popular beaches. This phenomenon was described in Australia by Maguire et al. [71] where "locals" would use less populated, but close-to-home, beaches as popular beaches were increasingly used by tourists. Thus, seasonal weather patterns and density-dependent changes in human behavior likely explain much of the variation in coast-wide water user activity across southern California.

Southern California beaches were dominated by surfers (40 ± 27%) and waders (35 ± 23%) coast-wide (Fig 2). Surfing, globally, is a $3.6 billion industry [72] and ease of access and consistent surf conditions are a highly desired quality for board sport beaches [68], which accounts for the high ratios of these water user groups in this study. Swimmer abundance was likely underrepresented in this study, e.g., Kane et al. [73] found that swimmers at Australia beaches were most abundant later in the day. In our study, most surveys were conducted between 08:00 and 13:00 because of optimal flight conditions, thus there is bias towards under estimation of this group that were more likely ocean swimming off California beaches earlier in the morning or later in the afternoon and evening. While this group can still co-occur with JWS at

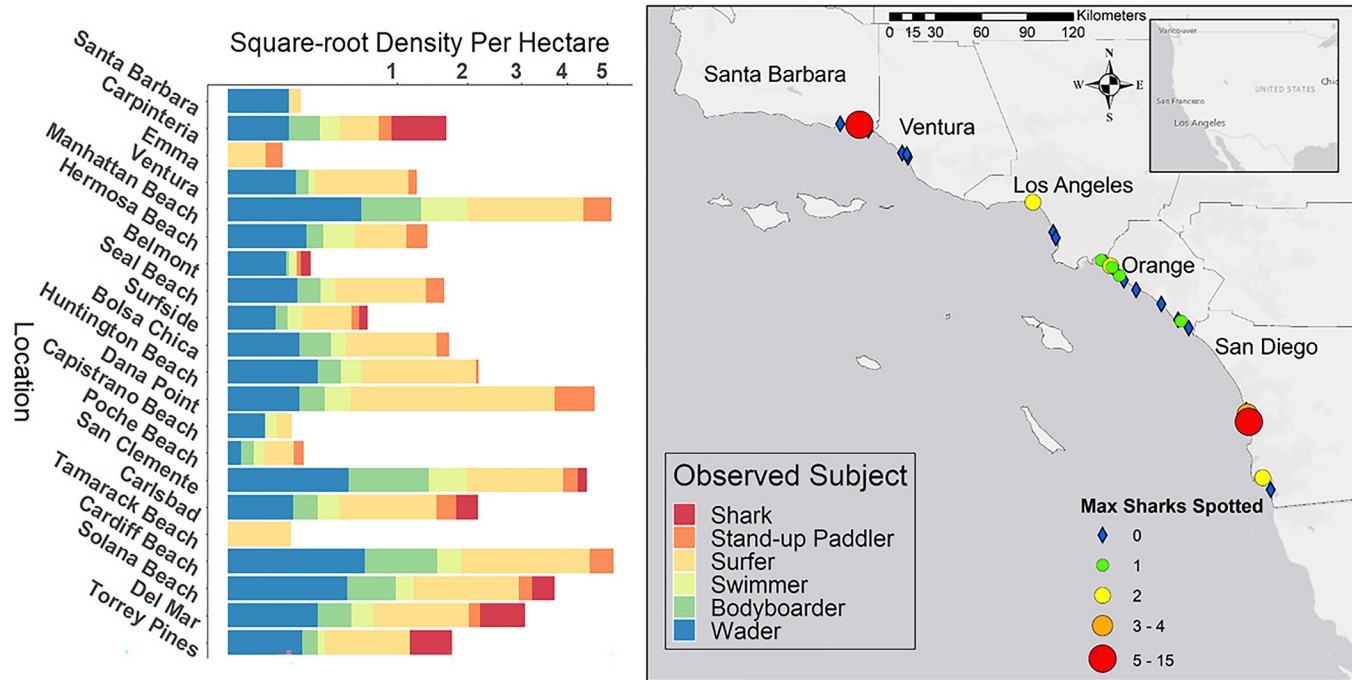

**Fig 7. Locations of juvenile white shark sightings during survey in southern California, and average observed subject density per survey day at each location.** Locations are organized from North to South. Base maps and map data were produced in ArcGIS. Base map data produced by Esri, Here Technologies, Garmin, OpenStreetMap contributors, General Bathymetric Chart of the Oceans, and National Oceanic and Atmospheric Administration National Geophysical Data Center. Map data from the USGS can be viewed here: https://apps.nationalmap.gov/viewer/.

high rates (Fig 2), they primarily use water closer to the wave break separating them from a majority of JWS activity (Figs 3, 5C and 6C). Furthermore, although they were observed to be the most abundant group, there is likely an underrepresentation of surfers in our study as well. Surfers may be exploiting better surfing waves at dawn and dusk, which are before and after surveys were conducted. To account for this, quantifying co-occurrence on a survey day basis instead of a "per survey" basis also may alleviate some bias in co-occurrence rate between JWS and these groups, as they were still observed across morning hours (08:00–13:00). Not surprisingly, coast-wide human water user abundance was also positively correlated with high UV index, SST, and negatively correlated with high total cloud cover and wind speeds (Fig 4). Freitas et al. [74] identified these environmental conditions as "ideal" for predicting high beach use satisfaction for beach tourism. Regardless of beach location, these environmental parameters are highly predictive of all water user activity coast-wide.

While human abundance along the beach is highly predictable, beach-specific human space use and nearshore distribution were also consistent across this study. The majority of water users across southern California, regardless of activity, were observed within 200 m of public beach access (S1 Fig). Due to the variation in public beach access across surveyed locations, human beach-specific distributions differed by location (Figs 5A and 6A). For example, in Carpinteria water user distribution area (50% KDE) was ~0.02 km$^2$, while in Del Mar water users were observed over ~0.44 km$^2$ of nearshore habitat across the entire study period (Figs 5A and 6A). Despite a wide variation in environmental factors among individual survey days people were rarely, if ever, observed in areas outside of these observed recreation hotspots (Figs 5 and 6). Kane et al. [73] and Da Silva et al. [75] found that beach goers often group around public beach access points and there was an exponentially negative relationship between human density along beaches and distance from public car parks [76]. Although, it is important to note that due to Federal Aviation Administration flight restrictions, all the beaches included in this study were standard, easily accessible beaches. As a result, this may bias the distance from public beach access, as experienced surfers may seek better "surfing" waves that are located along cliff sides, or away from easy public access. However, public beach access combined with marine nearshore structure (e.g., patch reef, permanent sandbars, estuary outflow) that create consistent waves for human water use likely explain why environmental parameters affected nearshore abundance but not beach-specific spatial distributions. For example, wave height may have affected how many surfers were at a certain beach, but beach-specific spatial distribution rarely changes. Furthermore, Del Mar Beach likely attracted more people due to more dispersed beach access, better wave conditions, a more densely populated urban area surrounding the beach, and higher dispersion of people due to higher human use. Waders were least likely to encounter JWS, whereas surfers, swimmers, and SUPs had the highest likelihood of encountering sharks based on their nearshore distributions at either aggregation site, despite slight differences in overall distribution (Figs 5C and 6C). Even though JWS were rarely seen outside of aggregation sites, water user nearshore distribution patterns were generally consistent, with slight variation from beach to beach, across all locations (Fig 3). This study indicates that human water user activity in southern California is predictable across coast-wide, beach-specific, and nearshore distributions despite lack of predictability in JWS aggregation formation.

### Shark distribution along southern California

Coast-wide JWS distribution was clumped and only observed at 8 of the 26 surveyed locations. From 2020 through 2021, JWS were primarily observed at two beach locations: Carpinteria and Del Mar. Previous and on-going acoustic telemetry and satellite tracking studies indicate

JWS form loose aggregation (< 40 individuals) at particular beach locations, exhibiting high site fidelity to these locations (~ 7 km$^2$) for periods up to months, although these aggregation hot spot locations change every few years [28,30,33,35,77]. This clumped distribution could be explained by individuals seeking preferred environmental conditions, increased forage base, and social benefits [28,33,35]. Based on size distribution (< 3 m TL), density, and site fidelity of sharks observed in these areas, these areas are likely serve as nursery habitat [78].

JWS were rarely seen during surveys outside of aggregation sites, and those observed were detected farther from the shoreline, with a median distance from the wave break of 216 m at locations outside of aggregation sites compared to a median of 86 m at aggregation sites. Lower detection rates of sharks outside aggregation sites may have been attributed to a lack of surveys within restricted airspaces (e.g., Santa Monica Bay), fewer sharks transiting between aggregation sites, or sharks occurring at very lower densities at surveyed beaches outside of aggregation sites (Fig 1). Since aerial surveys in this study only extended to 500 m from the shoreline, it is possible that some JWS were further offshore during beach surveys, and thus, not observed. Anderson et al. [35], found that JWS traveled quickly and directly after leaving one aggregation site for another. Young white sharks in Australia were also found to migrate between aggregation areas transiting quickly following the 60–120 m isobath [79]. Satellite tracking data has also indicated that some JWS in southern California may be using neritic waters out to 10–20 km offshore [32], and with individuals tracked up to 100 km off the coast of New York [45].

Within aggregation sites, there may also be bias and underrepresentation in shark observations, as sharks may be present but too deep to detect via aerial observation. This is an inherent limitation in using aerial surveys as a detection tool. However, Anderson et al. [35] observed that acoustically tagged sharks with pressure sensing transmitters tracked over the course of 9 months at Carpinteria had the vast majority of their detections within the top 3 m of the water column. This is within observable depths via drone or aerial surveys [43,51,80]. Furthermore, they also observed that JWS at this aggregation site were closer to the surface over the course of the day, and in particular between 08:00 to 13:00 [35]. This aligns with the timeframe when the majority of our surveys were conducted. It can be inferred that, while some JWS may have remained undetected, if JWS were present it is likely that they would be detected throughout the course of a survey day.

Because sharks and water users were consistently observed at aggregation sites, and based on their nearshore water user distributions, there is high likelihood that JWS are encountering water users frequently each day. JWS were predominantly within 100 m of the wave break (Figs 3, 5C and 6C) and well within human nearshore distribution ranges. In addition, certain environmental variables, e.g., high UV, summer months, wave height of ~1 m, or generally "good" weather, predicted increased water user activity as well shark abundance at aggregation sites, further increasing likelihood of shark-human encounters under those conditions (Fig 4). While aerial survey data indicate that JWS nearshore habitat use was much larger than that observed for water users at the same beaches, there was high spatial overlap between the two species (Figs 5A, 5B, 6A and 6B). This indicates that high spatial habitat overlap occurs year-round, depending on whether JWS overwinter. Previous tracking studies of JWS in southern California has indicated that YOYs and smaller juveniles would migrate during late Fall to Baja Vizcaino Bay, Baja, Mexico [29–32,77]. In recent years, there has been higher interannual variability in winter migratory behavior, with a slightly higher frequency of larger JWS over-wintering off the southern California nursery habitat in 2020 than 2019. While it is possible that JWS were not observed due to poorer winter flight conditions, a concurrent acoustic telemetry study by Spurgeon et al. [60], observed sharks returned to Carpinteria in late November 2020 after local temperatures dropped below an observed thermal threshold of

12˚C. As southern California waters continue to increase in temperature due to global climate change this low thermal threshold may not occur as often as it has in the past, further increasing the rate of overwintering by JWS. However, on-going coast-wide acoustic telemetry tracking of JWS in this area may further elucidate winter emigration and residency change over time due to interannual environmental conditions. While human-JWS distribution overlap is not consistent across the southern California coastline, the high degree of spatial overlap at aggregation sites does not appear to negatively affect beach safety or JWS survivability.

## Impacts of human-JWS overlap

Increasing popularity of human ocean recreational activities and the effects of climate change (hotter, drier summer weather [81]) may drive increases in nearshore ocean use, particularly along southern California. In addition, successful conservation efforts and improved fisheries management has allowed for recovery of white sharks along the California coastline [10–12], where extensive JWS nearshore nurseries have formed, resulting in growing spatio-temporal overlap in habitat use between humans and white sharks. Despite these rising trends, there is little evidence of increased frequency shark bites on humans in southern California. Over the 2-year survey period, only one minor potential unprovoked shark bite was reported across southern California at one of the aggregation sites. In spring of 2020 a swimmer was reportedly bitten by a marine animal with minor lacerations and stated she saw a JWS leaving the area. However, the injury could not unequivocally be identified as the result of a JWS bite. An aerial survey study conducted at La Reunion Island, a region known for high frequency of shark bites on people, found that areas of high shark-water user overlap did not result in increased shark bites on water users [26]. In addition, the probability of unprovoked shark bites across California was estimated to be extremely low [23]. Since 1950 there have been 130 reported white shark bites in California, with only 20 unprovoked white shark bites in southern California since 2000 [36,37].

While rare, surfers are bitten by sharks more frequently than any other recreational water user group [24,27,82], potentially due to relatively higher frequency of encounter and the potential for being mistaken for prey [83]. JWS at southern California aggregation sites were observed to co-occur with surfers more frequently than any group (54 out of 76 survey days co-occurrence was observed), had the highest abundance on survey days with co-occurrence (44 surfers present on average) and comprised 40 ± 27% of the average relative abundance of all water users at all beaches; yet there were no reported incidents involving shark-related injuries during this study. One explanation for this lack of negative interaction despite high co-occurrence may be attributed to YOY and JWS (1.5–3 m) feeding predominantly on benthic elasmobranchs, small pelagic schooling teleosts, and reef-associated teleosts [84–87]. These prey items differ considerably in size, shape, movement patterns, and are much smaller than humans. Since many of these JWS are about the same size or smaller than most surfboards, sharks may focus most of their attention on the benthos for foraging and may avoid or ignore surface oriented human water users. Shark are inherently risk averse when confronted with objects or animals of similar size [88]. Therefore, a surfboard/SUP (generally 1.8 m– 3.4 m) may discourage direct confrontation from a juvenile white shark (1.5 m– 3.0 m) and inherently reduce bite risk. While larger sub-adult white sharks (3.0–3.6 m male, 3.0–4.5 m female) exhibit an ontogenetic dietary shift from piscivores to an inclusion of marine mammals [84,86], they are thought to be more likely to mistake a surfer for a larger prey item such as pinnipeds which are common along the California coastline. Nonetheless, sub-adult white sharks were rarely observed along southern California beaches during surveys and based on historic fisheries data [9]. Lastly, Carpinteria and Del Mar are within the Northeastern Pacific

population of white shark's nursery habitat [28,32,89], where JWS may select these shallow water nearshore sites to reduce interactions with larger predators (e.g., adult white sharks, orca), and take advantage of warmer water conditions and abundant small prey juveniles to enhance growth rates [78].

Increasing JWS abundance at local beaches has raised human safety concerns, as well as protection of these sharks. For example, high human water activity may discourage JWS use of certain beaches due to direct disturbance of sharks or their prey. Other predators are known to avoid areas with high human activity despite high spatio-temporal overlap [90–96]. The two JWS aggregation sites surveyed during this study showed clear differences in the amount of water user activity relative to shark abundance. Carpinteria had a comparatively lower water user activity for most of the year, with notable increases during the Summer, whereas Del Mar, however, had comparatively higher year-round water user activity, with a higher abundance of surfers. Interestingly, we observed very little difference in JWS distribution between the two sites suggesting that water user activity may have little effect on JWS distribution (Figs 5 and 6). Lastly, although human abundance and beach use in southern California was consistent across years, JWS aggregation sites are ephemeral and shift every few years [28]. Benthic foraging habits of JWS and the predominantly surface orientation of human water users may explain why there were few differences observed in JWS abundance at heavily and less heavily used beaches where aggregations were monitored. Thus, it is possible, since humans were predominantly unaware of JWS activity in these areas, and were not intentionally interacting with them (e.g., chasing or harassing), that despite high co-occurrence and shared habitat use, risk of bites by JWS is low.

## Conclusions and impacts

This is one of the first studies to simultaneously quantify human and shark spatiotemporal distribution overlap over multiple spatial resolutions. Although other studies have focused on human or shark nearshore distribution or presence using drones, this study combines those methods to address risk of shark bites in southern California. Overall, human-JWS overlap is limited on a coast-wide basis; human-JWS only co-occur at areas where JWS aggregations have formed. In the nearshore, only groups that are outside the wave break overlap with JWS, who spend most of their time within 100 m of shore. Surfers are most likely to co-occur with JWS based on a high mean relative abundance within southern California and nearshore distribution overlap with JWS. Beach-specific human distribution is dependent on distribution of public beach access, while JWS beach-specific distribution is variable among beaches and harder to predict. As a result, human beach-specific area use may drive spatio-temporal overlap at specific beaches. However, while there is high, year-round spatio-temporal human-JWS overlap, no aggressive behaviors were observed nor unprovoked bites at aggregation sites in southern California during the study period. Thus, under current conditions, JWS bite risk on nearshore water users remains extremely low, despite high daily encounter rates at aggregation sites.

There are several potentially widespread impacts of this study. First, the methods used in this study could be readily adopted by California lifeguard agencies to monitor their beaches for shark activity. The New South Wales territory in Australia has developed a Shark Smart program that surveys local beaches in a similar fashion [46,97,98]. Using these methods, they can identify hotspots of shark activity at their beaches. Furthermore, instead of closing entire beaches, public safety officials may elect to restricting public beach access points to limit human-shark overlap at shark area use focal points. However, results from this study can also be used to justify not limiting beach access or closing beaches, which are known to have

economic impacts on local communities, because sharks under 3 m TL may not present an immediate danger to humans. Lastly, there are other populations and shark species that spend large amounts of time close to shore and close to humans, and further studies with similar methods could identify how other species may differ in nearshore behavior and distribution with respect to water users. Ultimately, more studies of this type at other locations could provide data needed to evaluate actual shark risk, thereby potentially eliminating the need for shark control or mitigation measures that may prove harmful to other unintended wildlife and the ecosystem.

## Supporting information

**S1 Fig. Histogram of observed human distance from closest public beach access.** The solid line defines the mean of the data, while the dashed line defines the median of the data.
(TIF)

**S2 Fig.**
(PNG)

**S1 Table. Source and method of data collection for all environmental variables included in this study.** All data was pulled from publicly available, nationally managed databases defined within this table.
(CSV)

**S2 Table. Pair-wise statistical comparison and output of the study-wide Kruskal-Wallis test results.** "NS" defines no significant difference between distributions, and $p < 0.05$ indicates statistically different distributions.
(CSV)

**S3 Table. Pair-wise statistical comparison and output of the Carpinteria Kruskal-Wallis test results.** "NS" defines no significant difference between distributions, and $p < 0.05$ indicates statistically different distributions.
(CSV)

**S4 Table. Pair-wise statistical comparison and output of the Del Mar Kruskal-Wallis test results.** "NS" defines no significant difference between distributions, and $p < 0.05$ indicates statistically different distributions.
(CSV)

**S5 Table. Survey image overlap between subsequent images based on altitude.** The discrepancy between 27.5 and 30.0 distance covered is the minimum speed and maximum survey speed to encompass the minimum and maximum image overlap.
(CSV)

**S6 Table. The resulting output of the final model derived from step-wise GAMMs for waders.** Values of $p < 0.05$ indicate significant effects on the response, presence during surveys.
(CSV)

**S7 Table. The resulting output of the final model derived from step-wise GAMMs for bodyboarders.** Values of $p < 0.05$ indicate significant effects on the response, presence during surveys.
(CSV)

**S8 Table. The resulting output of the final model derived from step-wise GAMMs for swimmers.** Values of $p < 0.05$ indicate significant effects on the response, presence during

surveys.
(CSV)

**S9 Table. The resulting output of the final model derived from step-wise GAMMs for surfers.** Values of $p < 0.05$ indicate significant effects on the response, presence during surveys.
(CSV)

**S10 Table. The resulting output of the final model derived from step-wise GAMMs for Stand-up Paddlers.** Values of $p < 0.05$ indicate significant effects on the response, presence during surveys.
(CSV)

**S11 Table. The resulting output of the final model derived from step-wise GAMMs for JWS.** Values of $p < 0.05$ indicate significant effects on the response, presence during surveys.
(CSV)

# Acknowledgments

Assistance with field surveys and image processing was provided by Haylee Kramer, Abby Henderson, Bailey Bonham, and Becca Prezgay. California State University Long Beach graduate students Taylor Smith, and Yamilla Samara provided visual observation for drone surveys. California State University Long Beach alumni Echelle Duffield and Emily Meese provided assistance in methodology planning. Dr. James Anderson provided methodological, and data analysis support. Citizen scientists Craig Prater, Carlos Guana, and Daina Buchner flew exploratory surveys. Considerable logistic support was provided by City, County and State of California Lifeguard Agencies for field surveys. Mark Rathsam and the city of Del Mar lifeguards in particular provided ample support.

# Author Contributions

**Conceptualization:** Patrick T. Rex, Jack H. May, III, Erin K. Pierce, Christopher G. Lowe.

**Data curation:** Patrick T. Rex, Erin K. Pierce.

**Formal analysis:** Patrick T. Rex.

**Funding acquisition:** Patrick T. Rex, Christopher G. Lowe.

**Investigation:** Patrick T. Rex, Jack H. May, III, Erin K. Pierce.

**Methodology:** Patrick T. Rex, Jack H. May, III, Erin K. Pierce.

**Project administration:** Christopher G. Lowe.

**Resources:** Christopher G. Lowe.

**Supervision:** Christopher G. Lowe.

**Validation:** Patrick T. Rex, Christopher G. Lowe.

**Visualization:** Patrick T. Rex.

**Writing – original draft:** Patrick T. Rex.

**Writing – review & editing:** Patrick T. Rex, Jack H. May, III, Erin K. Pierce, Christopher G. Lowe.

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
