## [Decision Letter · Decision Letter 0]

27 Mar 2023

PONE-D-22-35225Patterns of overlapping habitat use of juvenile white shark and human recreational water users along southern California beachesPLOS ONE

Dear Dr. Rex,

Thank you for submitting your manuscript to PLOS ONE. After careful consideration, we feel that it has merit but does not fully meet PLOS ONE’s publication criteria as it currently stands. Therefore, we invite you to submit a revised version of the manuscript that addresses the points raised during the review process.

Annotated on a pdf of the manuscript, the reviewer provided some helpful comments and questions that need to be addressed.

We look forward to receiving your revised manuscript.

Kind regards,

John A. B. Claydon, Ph.D.

Academic Editor

PLOS ONE

Journal Requirements:

5. We note that Figures 1, 5, 6 and 7 in your submission contain map/satellite images which may be copyrighted. All PLOS content is published under the Creative Commons Attribution License (CC BY 4.0), which means that the manuscript, images, and Supporting Information files will be freely available online, and any third party is permitted to access, download, copy, distribute, and use these materials in any way, even commercially, with proper attribution. For these reasons, we cannot publish previously copyrighted maps or satellite images created using proprietary data, such as Google software (Google Maps, Street View, and Earth). For more information, see our copyright guidelines: http://journals.plos.org/plosone/s/licenses-and-copyright.

(1) You may seek permission from the original copyright holder of Figures 1, 5, 6 and 7 to publish the content specifically under the CC BY 4.0 license.  

7. Please upload a copy of Supporting Information Figure 1 which you refer to in your text on page 35.

Reviewers' comments:

Reviewer's Responses to Questions

**Comments to the Author**

1. Is the manuscript technically sound, and do the data support the conclusions?

Reviewer #1: Yes

2. Has the statistical analysis been performed appropriately and rigorously? 

Reviewer #1: Yes

3. Have the authors made all data underlying the findings in their manuscript fully available?

Reviewer #1: No

4. Is the manuscript presented in an intelligible fashion and written in standard English?

Reviewer #1: Yes

5. Review Comments to the Author

Reviewer #1: This is a well-written manuscript providing valuable information about spatio-temporal human-shark overlap in Southern California. I only had mostly minor comments and queries (see attached pdf) which the authors should be able to address easily.

Well done for this interesting study.

6. PLOS authors have the option to publish the peer review history of their article (what does this mean?). If published, this will include your full peer review and any attached files.

Reviewer #1: **Yes: **Charlie Huveneers

---

## [Author Response · Author response to Decision Letter 0]

9 May 2023

Here is our full response to the reviewer and editor comments. This is also in our "Response to the Reviewers" file uploaded in this document.

Here, again, is our statement on the maps used that were formatted to use open-sourced map options: We have also fixed Fig5 and Fig6 to include open sourced map tiles from Stamen Maps. Their statement on map tile use can be found here: https://stamen.com/open-source/#:~:text=For%2020%20years%2C%20Stamen%20Design,our%20publicly%20available%20map%20tiles. 

The authors thank Dr. Claydon, reviewers, and PLoS One staff for their constructive reviews of this paper. We have compiled all the reviewer comments and journal submission requirements into this document for ease of review. We have included line numbers for where comments occurred. Our responses and comments are in bold.

Reviewer 1:

We greatly appreciate the comments and positive feedback from the reviewer and editor. We have done our best to address all comments and suggestions in improving the manuscript.

Line 28-29. Considering the non-homogeneous distribution of the JWS, it might be worth specifying whether the drone flights were homogeneous across the Californian coastline or if they were also mostly around Carpinteria and Del Mar?

-Modified to the sentence to try to clear up the rationale and spatial distribution of flights along the coast. 

Line 31. Do not use acronym when starting a sentence.

I'd also consider if this acronym is necessary and I'd recommend to use juvenile white shark in full.

-Agreed, the acronym was used to reach abstract limit. The abstract has been reworded to be below 300 word abstract limit and have removed all acronyms. 

Line 32-33. See above comment, how many drone flights were at these two beaches? 

-We agree that the issue of effort between aggregation sites and non-aggregation sites needs to be addressed. However, because of space limitations we do not believe the abstract is the appropriate place to address this, so we have added clarification in the Methods section in the paragraph before “Survey Processing” to address this comment. 

Line 38-39. Specify what you are basing this on (I'm assuming that this is based on the low numer of bites?)

-The text has been fixed to address this point.

Line 39-40. There are many other reasons that juv. white shark might not be biting people at these locations. Only mentioning one of those and suggesting that this is the reason is misleading and undermines all the other possible reasons. This is particularly true as this is speculative.

Other possible reasons are:

1) juv. white sharks were never interested in humans (i.e. no habituation was required) 

2) humans are seen at potential threats and juv. white sharks rather avoid direct contact

Etc....

-We agree with the reviewer, we have removed habituation from the abstract. We have moved discussion of possible other explanations for low bite risk to the discussion. 

Line 83-86. While this might be true, and that there is an overlap between the size class of shark responsible for bites and that of the juv. white sharks forming the Californian aggregation, what is the size distribution frequency of the Californian aggregation?

-Unfortunately, no published data to date has fully described the size frequency distribution of the Northeastern Pacific population to address the reviewer’s comment here. We have cited Ugoretz’s 2022 paper on unprovoked shark bites in California that at least describes the size of sharks involved in bite incidents in California to corroborate this statement.

Line 152-153. If you measured distance from every subject and had such high overlap, you would have measured the distance of many subjects twice? How did you account for this?

-High overlap was built into the methods so that each individual would be observed and accounted for. Most beaches were sparsely populated by humans, which helped track and avoid duplicate measurements, of humans within the first transect of the survey. People were easy to track, even at high altitudes, due to differences in body shape/size, clothing, and board size/color. We also erred on the side of underestimation, by not including any human measurements if there was a potential risk of double-measuring. We were not able to account for multiple observations of the same shark individual. However, to account for this discrepancy, both humans and sharks only had 1 GPS and distance measurement conducted at the instant the subject was closest to the center of the image. 

-This clarification has been added to the text

Line 205-209. This is a relatively coarse way of comparing human shark distance across subject groups. A GLM might have been more robust and provided more flexible analysis.

-Unfortunately, the distribution of the data (distances from the wave break) was highly irregular across the entire dataset (See Figure 3 for example). Normalization techniques did not reduce heteroscedastic distributions of the data. While this is a coarse analytical approach, we felt this was the best way to analyze the data without drastically violating GLM assumptions. 

-A sentence has been added to reflect these decisions in the text.

Line 209-210. Was there no seasonality worth investigating?

-This is a very good point. Since sharks were only observed at the aggregation sites referred to within this study, we felt it was inappropriate and misleading to assess mean relative abundance by season at the coast-wide level; many sites only had sharks observed in one season then never again. Mean relative abundance here is meant to assess how human use of similar beaches, such as Carpinteria and Del Mar, may affect the rate of co-occurrence and potential encounters. Seasonality is better described in Figures 5 and 6 at specific aggregation sites.

-a sentence has been added to the text to reflect these decisions.

Line 221-226. It might be interesting to compare this to another definition of human-shark co-occurrence to test how much this definition affects your results. e.g. would you have the same results if you defined co-occurrence as 'whenever sharks were spotted within 100 m of humans'?

-This is a great point by the reviewer. Due to high variability, there very well may be large discrepancies when sharks were observed at the same beach, and when they were within the same frame of the drone even. This would bring them within a maximum distance of 210 m using the drone during this study. However, this was kept intentionally broad within the scope of this study. We were interested in investigating the environmental parameters and locations that increased or decreased common spatio-temporal use of the same location across the entire coast of California. We are currently investigating the phenomenon of when sharks were spotted within 100 m of humans and how they are/if they are affected by these same parameters. We opportunistically observed over 300 instances of this over the course of this study, but again that is outside the scope of this paper. 

-A sentence has been added to the text to reflect this decision

Line 221. unnecessary space

-Thank you. Removed. 

Line 223-224. how did you test these correlations and what threshold did you use?

-We apologize for the lack of clarity. A correlation matrix was used to assess correlation with associated p values to assess significance of correlation. This was used to reduce model complexity with redundant variables, but this was not clear. Correlation within this sentence was poor wording.

-We have added threshold for significance within the correlation matrix and clarification of redundancy in the text.

Line 236-238. If this is the case, the correlation test between these variables should be lower than for the other correlated variables.

-We have added clarification within the text to address this comment.

Line 301. Might be worth adding something like: "and 58 (78.4%) of those were at aggregation areas."

-We have included the suggested phrase into the text. 

Line 352-353. Where is the data showing this?

-This is poor wording and has been modified in the text. We have re-run and re-assessed our models to check significance, and Location is not significant (SI table 11). While sharks were generally only observed in 2 locations, they were only occasionally observed in other locations along the coastline which may explain why this variable was not significant. In a biological sense, this makes sense. We still cannot predict where JWS will aggregate, so it follows that location would not be significant within this study.

-The text has been modified to reflect this. 

Line 354-354. Seems like a result you already stated

-Agreed, we have removed the statement from the text

Line 362-363. So, you would have expected that cloud cover/base height would have had some impact on shark abundance. Strange that it didn't UV level did. 

I'm not sure how you checked for correlations, but it might be that these were correlated?

-These are significantly correlated (p < 0.05), and this sentence was a potential explanation why they were not significant via analysis. There is also an inherent limitation in the ability to fly at low cloud cover via FAA regulations (cloud base height must be above 610 m) which may have also limited the dataset. 

-This has been added to the text.

Line 386-387. From comparing Fig 3 and 5c, it doesn't seem like the juv. white shark distribution was that different between coast-wide distribution and Carpinteria-specific distribution

-We believe the figures were uploaded improperly. There are different distributions and different statistical overlap between coast-wide and the two aggregation sites. We have added both figures at the end of this response for ease of comparison. We will also re-upload these figures into the PLoS One author portal upon re-submission to ensure this mistake does not make it into the final article. 

Line 392-393. How did you have 38 survey days out of 26 months of surveying, with surveys undertaken once a month (according to your methods)?

-JWS positions and measurements were also included opportunistically when CSULB scientists were tagging free swimming JWS. Human surveys were not included in these activities to limit the discrepancy in effort between sites. However, co-occurrence was still observed if sharks swam next to human subjects even if human surveys were not conducted. 

-Clarification has been added to the methods section. 

Line 468-470. Similarly, surfing distribution/patterns might also have been biased as surfers are more likely to use good breaks early mornings/late afternoon.

-This is a good point by the reviewer. We have added this to the text

Line 486-487. Beach access might be important for waders, swimmers, SUP, but surfers (especially experienced one) are less likely to use standard beach access and therefore more likely to be at different beaches than swimmers

-Agreed. We have added text to address this comment and the following one at the same time. 

Line 491-493. Is this biased by the beaches surveys? For example, if you surveyed areas like Trestles, you'd see a very different patterns with surfers in high density quite far from public access. You need to more clearly specify that this is for standard easily accessible public beaches and doesn't apply to the whole coast.

-Agreed. Due to FAA restrictions on site location based on restricted airspace, all the beaches included in the study are standard, easily accessible public beaches. This may bias may be why we observed this phenomenon. 

-We have added text to address this and the above comment.

Line 506-507. but in parts of your results, you seem to suggest that SUP distribution at Carpenteria was different to coast-wide?

-We have added text to clarify

Line 521-523. Apologies if I've missed it, but is this in the results section?

-We respectfully point the reviewer to lines 275-278 (in the original text) where this is mentioned with statistical test results. 

Line 538. vague. Which ones?

-Added text to clarify

Line 567. what do you mean by this?

-A woman was bitten in Spring of 2020 by a marine animal, but it could not un-equivocally be identified as a JWS. Clarification has been added to the text.

Line 571-573. See recent paper in scientific report going through shark bites in California.

-This citation has been included.

Line 581-582. I agree with what is being said in this paragraph, but would recommend to present and discuss other possible reasons for the lack of bites/negative interactions in a bit more details. 

See earlier comment in the abstract.

This paragraph is also not consistent with the abstract, which refers to habituation while this paragraph doesn't mention it.

-Habituation has been removed from the abstract, as we agree with the reviewer that this may not be appropriate within this paper's scope. 

-Added text explicitly referencing sharks, as predators, being risk averse, and size being a deterrent to direct confrontation. Also added text that alludes to the fact that this is a nursery habitat which means JWS may be ignoring people in general to focus on growth. 

Line 601. be careful with the wording here. You didn't assess behavior so can't talk about differences in behavior between the two aggregation area. Should be 'distribution' or something similar.

-Agreed, we have substituted the suggestion into the text

Line 609-610. or never worried about humans... Without any data about this, the authors should be careful not to make such statement (or at least present all/other possible explanations)

-Agreed. This was speculative and has been reworded. 

Line 617-618. My only major comment of this manuscript relates to juv. white shark distribution/distance from shore. What is the bathymetry from the wave break to ~500 m from shore and how consistent is shark depth use across this gradient? Since you're using aerial surveys, you're obviously relying on sharks being close to the surface to detect them. I'm assuming at water depth 500 m from the shore is deeper than by the wave breaks, so is it possible that sharks are as, or more, abundant further from shore but are deeper and less detectable by drones? This would not impact human-shark overlap as humans distribution remains close to shore, but it could impact the shark distribution. The Discussion should acknowledge and discuss this possible bias, and whether it affects (or not) the findings/results. Or you could use your group's acoustic tracking data to show that the drone data is consistent with the acoustic tracking data and shows primary use of nearshore areas

-We thank the reviewer for bringing attention to this. This is a major caveat of using drones as detection tools, there is an inherent limitation in the technology. While no technology will ever detect every single animal in the area, e.g. acoustic telemetry is limited to receiver location and tagged animals, there may be bias present that should be discussed. We have added an explanation for why we believe this bias is limited and included a citation from an acoustic study performed in the same area. 

-Text has been modified to reflect this in the second paragraph in the discussion under “Shark distribution along southern California”.

Line 629-633. There are many references from NSW which describe this program and use of drones as shark bite mitigation measure which should be cited here.

-We have added citations in reference to this into the text.

Journal Submission Requirements that need to be addressed: 

-Thank you for bringing this to our attention. Headings have been formatted properly, and title page has been adjust to follow PLOS ONE style templates.

-Thank you for bringing this to our attention. We have added a brief statement on field site access. We have also included our IACUC protocol number for JWS although no animals were handled within this study. 

-Thank you for bringing this to our attention, we will correct this issue on resubmission. 

-Raw data needed to repeat these observations have been uploaded to Center for Open Science Repository. This can be found here: https://osf.io/esvjq/, and will be made public upon acceptance. 

5. We note that Figures 1, 5, 6 and 7 in your submission contain map/satellite images which may be copyrighted. All PLOS content is published under the Creative Commons Attribution License (CC BY 4.0), which means that the manuscript, images, and Supporting Information files will be freely available online, and any third party is permitted to access, download, copy, distribute, and use these materials in any way, even commercially, with proper attribution. For these reasons, we cannot publish previously copyrighted maps or satellite images created using proprietary data, such as Google software (Google Maps, Street View, and Earth). For more information, see our copyright guidelines: http://journals.plos.org/plosone/s/licenses-and-copyright.

(1) You may seek permission from the original copyright holder of Figures 1, 5, 6 and 7 to publish the content specifically under the CC BY 4.0 license. 

-Thank you for bringing this to our attention. We have added appropriate citations within the figures as required by the TOS for both google maps and ArcGIS.

-Thank you for bringing this to our attention, we have added in SI captions at the end of the manuscript after the References section. 

7. Please upload a copy of Supporting Information Figure 1 which you refer to in your text on page 35.

-Thank you for bringing this to our attention, we have added in SI Fig 1 at the end of the manuscript and will submit the figure to PACE to ensure it fits journal submission requirements. 

-Thank you for bringing this to our attention. Unfortunately, we were using a reference tool that was pulling in references incorrectly. All references have been corrected.

---

## [Editor Report · Decision Letter 1]

19 May 2023

Patterns of overlapping habitat use of juvenile white shark and human recreational water users along southern California beaches

PONE-D-22-35225R1

Dear Dr. Rex,

We’re pleased to inform you that your manuscript has been judged scientifically suitable for publication and will be formally accepted for publication once it meets all outstanding technical requirements.

Kind regards,

John A. B. Claydon, Ph.D.

Academic Editor

PLOS ONE
---

## [Editor Report · Acceptance letter]

26 May 2023

PONE-D-22-35225R1 

Patterns of overlapping habitat use of juvenile white shark and human recreational water users along southern California beaches 

Dear Dr. Rex:

I'm pleased to inform you that your manuscript has been deemed suitable for publication in PLOS ONE. Congratulations! Your manuscript is now with our production department. 

Kind regards, 

on behalf of

Dr. John A. B. Claydon 

Academic Editor

PLOS ONE